# Gait Training Using the Honda Walking Assistive Device^®^ in a Patient Who Underwent Total Hip Arthroplasty: A Single-Subject Study

**DOI:** 10.3390/medicina55030069

**Published:** 2019-03-14

**Authors:** Kazunori Koseki, Hirotaka Mutsuzaki, Kenichi Yoshikawa, Yusuke Endo, Takayuki Maezawa, Hanako Takano, Arito Yozu, Yutaka Kohno

**Affiliations:** 1Department of Physical Therapy, Ibaraki Prefectural University of Health Sciences Hospital, 4733 Ami, Inashiki-gun, Ibaraki 300-0331, Japan; koseki@ami.ipu.ac.jp (K.K.); yoshikawak@ami.ipu.ac.jp (K.Y.); maesawat@ami.ipu.ac.jp (T.M.); takanoh@ami.ipu.ac.jp (H.T.); 2Center for Medical Sciences, Ibaraki Prefectural University of Health Sciences, 4669-2 Ami, Inashiki-gun, Ibaraki 300-0394, Japan; yodua@ipu.ac.jp (A.Y.); kohno@ipu.ac.jp (Y.K.); 3Department of Orthopaedic Surgery, Ibaraki Prefectural University of Health Sciences Hospital, 4733 Ami, Inashiki-gun, Ibaraki 300-0331, Japan; 4Department of Physical Therapy, Ibaraki Prefectural University of Health Sciences, 4669-2 Ami, Inashiki-gun, Ibaraki 300-0394, Japan; endoy@ipu.ac.jp; 5Department of Rehabilitation, Ibaraki Prefectural University of Health Sciences Hospital, 4733 Ami, Inashiki-gun, Ibaraki 300-0331, Japan; 6Department of Neurology, Ibaraki Prefectural University of Health Sciences Hospital, 4733 Ami, Inashiki-gun, Ibaraki 300-0331, Japan

**Keywords:** total hip arthroplasty, gait training, The Honda Walking Assistive Device^®^

## Abstract

*Background and objectives:* The Honda Walking Assistive device^®^ (HWA) is a light and easy wearable robot device for gait training, which assists patients’ hip flexion and extension movements to guide hip joint movements during gait. However, the safety and feasibility of robot-assisted gait training after total hip arthroplasty (THA) remains unclear. Thus, we aimed to evaluate the safety and feasibility of this gait training intervention using HWA in a patient who underwent THA. *Materials and methods:* The patient was a 76-year-old woman with right hip osteoarthritis. Gait training using HWA was implemented for 20 sessions in total, five times per week from 1 week to 5 weeks after THA. Self-selected walking speed (SWS), step length (SL), cadence, timed up and go (TUG), range of motion (ROM) of hip extension, and hip abduction and extension torque were measured preoperatively, and at 1 (pre-HWA), 2, 3, 4, 5 (post-HWA), and 10 weeks (follow-up) after THA. The gait patterns at SWS without HWA were measured by using three-dimensional (3D) gait analysis and an integrated electromyogram (iEMG). *Results:* The patient completed 20 gait training sessions with no adverse event. Hip abduction torque at the operative side, hip extension torque, SWS, SL, and cadence were higher at post-HWA than at pre-HWA. In particular, SWS, TUG, and hip torque were remarkably increased 3 weeks after THA and improved to almost the same levels at follow-up. Maximum hip extension angle and hip ROM during gait were higher at post-HWA than at pre-HWA. Maximum and minimum anterior pelvic tilt angles were lower at post-HWA than at pre-HWA. The iEMG of the gluteus maximus and gluteus medius in the stance phase were lower at post-HWA than preoperatively and at pre-HWA. *Conclusions:* In this case, the gait training using HWA was safe and feasible, and could be effective for the early improvement of gait ability, hip function, and gait pattern after THA.

## 1. Introduction

Total hip arthroplasty (THA) is a well-accepted treatment for patients with severe hip osteoarthritis (OA) [1]. THA contributes to higher functional ability, better pain relief, and improved quality of life and social participation [2,3]. Although gait function and patterns improve following THA, the functional improvements, such as gait speed, gait symmetry, and gait kinetic and kinematic parameters, are slow [4,5,6,7]. Female patients with THA walked slower than age-matched female reference subjects under maximum and normal speed conditions at about 4 weeks [4], 8 weeks [8], and even at 6 months [6] post-surgery. Furthermore, they were not able to achieve normal gait kinematic parameters, such as pelvic tilt angle, maximum hip extension angle in the stance phase, and hip flexion angle in the swing phase, compared to normal elderly subjects [6,8,9,10,11].

Recently, robot-assisted gait training (RAGT) using Lokomat^®^ (Hocoma, Volketswil, Switzerland) and Hybrid Assistive Limb^®^ (Cyberdyne, Tsukuba, Japan) has been widely used by individuals with stroke [12], a spinal cord injury [13], cerebral palsy [14], and total knee arthroplasty [15]. Many of these devices are heavy and have a strong exoskeleton mechanism to provide lower leg support, because they are intended to be used during repetitive gait exercises for severely paralyzed patients. However, there are only a few studies on the effect of RAGT after THA.

The Honda Walking Assistive Device^®^ (HWA, Honda Motor Corporation, Tokyo, Japan) is a wearable robot device for gait training, has a simple mechanism, and is light weight and easy to wear. Actuators placed at the hip joint assist hip flexion and extension movements to guide left–right symmetry and hip joint movements during walking. The angular and torque sensors placed at the hip joint monitor the hip joint angles and the assist torque, which is the device assisting hip joints based on the results calculated by an algorithm for regulating correct walking.

Given that patients who underwent THA do not have severe paralysis, strong assistance and large-scale equipment are not necessary. Furthermore, in these patients, improvement of hip joint movements and gait symmetry during walking with this device is believed to be effective to improve their gait ability. Therefore, we aimed to examine the safety of the training intervention and evaluate the effect of training using HWA on gait ability, muscle strength, hip range of motion (ROM), pain, gait kinematics, and muscle activity during walking in a patient who underwent THA.

## 2. Materials and Methods

### 2.1. Case Presentation

The patient is a 76-year-old woman with right hip OA. Preoperatively, her hip ROMs were as follows (right (R)/left (L)): hip extension −10°/15°, hip flexion 90°/125°, and hip abduction 25°/45°. Hip muscle strength measured by Manual Muscle Testing preoperatively was as follows (R/L): hip extension 4/5 and hip abduction 4/5. Spinomalleolus distance (R/L) was 69.0/71.5 cm. She had difficulty in gait and in performing activities of daily living (ADL), such as changing socks, bathing, stair use, getting in or out of a car, and shopping, because of pain and stiffness of the right hip. Her gait had features of pelvic rotation and anterior pelvic tilt accompanied with pain and restriction of hip extension during the right stance phase. Therefore, she underwent THA. Conventional rehabilitation programs, such as sitting, standing, and gait training using a walker with a physical therapist, were performed at 1 day after surgery according to the clinical pathway in our hospital.

This study was approved by the Ibaraki Prefectural University of Health Sciences Ethics Committee (approval no. e192). Then, sufficient explanation regarding study procedures was provided to the patient prior to obtaining written consent for study participation.

### 2.2. HWA Intervention

Gait training using HWA started at 1 week and ended at 5 weeks after THA (Figure 1). HWA training was implemented for 20 sessions in total, five times per week in an in-hospital setting. Each HWA training intervention was performed for less than 20 min, excluding resting time. The assist torque was set for each intervention on the basis of the patient’s subjective comfort with walking and a gait assessment by the physical therapist to obtain optimal walking from the patient. In the first half of the intervention period, the physical therapist assisted the patient to suppress pelvic rotation during gait training. The patient did not use any crutches, canes, or walkers during HWA training. The patient also underwent conventional rehabilitation, including ROM exercises, stretching, muscle training of the hip joint, balance exercises, and ADL exercises (changing clothes, using the toilet, bathing, and so on) during the intervention period.

### 2.3. Outcome Measures

The intervention and measurement timing are shown in Figure 2.

The parameters, including walking ability, hip function, and gait kinetics and kinematics, were measured preoperatively and at pre-HWA (1 week after THA), post-HWA (5 weeks after THA), and follow-up (10 weeks after THA) (overall measurement). Self-selected walking speed (SWS), step length (SL), cadence at SWS, timed up and go (TUG), ROM of hip extension, and hip abduction and extension torque were also measured every week during the intervention period (a weekly measurement). The timing and kinds of measurements are shown in Figure 2.

Any adverse events, such as the appearance of hip pain and skin problems (redness or scarring due to contact with equipment) during gait training with HWA were carefully observed.

Measurements related to walking ability included SWS, SL, cadence at SWS, and TUG. SWS tests were carried out on a walking path with 3 m of spare path before and after the 10 m measurement section. To calculate SL and cadence, the number of steps during the SWS tests was counted. The average values of three trials were adopted as the SWS, SL, and cadence.

Measurements related to hip function include the ROM of hip extension, hip abductor, and extensor torque (as muscle strength of gluteus medius (GMed) and gluteus maximus (GMax)) and Western Ontario and McMaster Universities Osteoarthritis Index subscales of pain (WOMAC-p), stiffness (WOMAC-s), and physical function (WOMAC-f) scores. WOMAC was divided into the subscales pain (five items), stiffness (two items), and physical function (17 items). The scores were summed for the items in each subscale, with possible ranges as follows: pain = 0–20, stiffness = 0–8, and physical function = 0–68. ROM of hip extension was measured by a medical goniometer in 5° increments. Torque of the hip was measured by Micro-FET 2^®^ (Hoggan Scientific, Salt Lake City, UT, USA) in a supine position. The maximum joint abduction and extension torque were measured in three sets, and the average values of the three trials were adopted.

### 2.4. Gait Analysis Using Three-Dimensional (3D) Gait Analysis and EMG

The gait patterns at SWS without HWA were measured through a 3D gait analysis (Vicon Nexus, Oxford Metrics, Oxford, UK) and electromyogram (EMG) (Trigno Wireless Systems, Delsys Inc., Natick, MA, USA). A widely used conventional marker set model (Plug-In Gait Model, Oxford Metrics) was used to calculate the joint angles after the definition of segments. Kinematic data were collected at a sampling rate of 100 Hz. Kinematic peak joint angles and angle range at the anterior pelvic tilt, hip (flexion and extension), knee (extension and flexion), and ankle (dorsi- and plantarflexion) were measured.

EMG sensors were placed on the GMax, GMed, vastus lateralis (VL), vastus medialis (VM), biceps femoris (BF), tibial anterior (TA), and gastrocnemius medial head (GCm) on the right side of the patient’s leg. The attachment position of each electrode was in accordance with the method of Surface EMG for the Non-Invasive Assessment of Muscles (SENIAM). EMG data were collected at a sampling rate of 2000 Hz. Subsequently, an absolute EMG was integrated by time to calculate an integrated electromyogram (iEMG). To standardize the time of one gait cycle as 100%, all data were interpolated by the third spline.

## 3. Results

The patient completed 20 gait training sessions and subsequent measurements. No adverse event was observed during gait training with HWA.

### 3.1. Gait Ability and Hip Function Parameters

Table 1 shows the measurements related to walking ability and hip function during the interventions and measurement period. Although the ROM of hip extension decreased, and TUG increased temporarily, at 1 week after THA, TUG was lower after 2 weeks than at pre-HWA. Hip abduction torque at the operative side, hip extension torque, SWS, SL, and cadence were higher, whereas the WOMAC-p, WOMAC-s, and WOMAC-f scores were lower at post-HWA than at pre-HWA. In particular, SWS, TUG, and hip torque showed a remarkable increase at 3 weeks postoperatively and improved to almost the normal levels at 10 weeks postoperatively.

### 3.2. Gait Kinematic Parameters and iEMG

The gait kinematic parameters are summarized in Table 2 and Figure 3. The characteristics of the patient’s gait preoperatively were as follows: a lack of hip extension, excessive anterior pelvic tilt in the stance phase, and decreased hip ROM during gait. A reduction in maximum hip extension and increased anterior pelvic tilt were observed at pre-HWA. The maximum hip extension angle and hip ROM during gait had increased at post-HWA, but had slightly decreased at follow-up. The maximum and minimum anterior pelvic tilt angles had decreased at post-HWA, but had slightly increased at follow-up. The maximum knee flexion angle at pre-HWA, post-HWA, and follow-up had increased compared to that at the preoperative stage.

The iEMG of GMax, GMed, VL, VM, BF, TA, and GCm during SWS is summarized in Figure 4. The iEMG of GMax and GMed at pre-HWA was reduced compared to at the preoperative stage. Although the iEMG of GMax and GMed at post-HWA had further decreased, the iEMG of GMed had slightly increased at follow-up. The iEMG of VL, VM, and BF at pre-HWA, post-HWA, and follow-up was increased compared to at the preoperative stage. Although the iEMG of GCm had temporarily increased at pre-HWA, it had decreased at post-HWA and follow-up.

## 4. Discussion

In this paper, we investigated the safety, feasibility, and effectiveness of the new exoskeleton gait training robot HWA in a patient who underwent THA. The patient completed a total of 20 gait training sessions without any adverse events. It was considered that HWA could be operated safely and feasibly after THA. The postoperative WOMAC score was remarkably improved, indicating that improvement in pain relief, hip joint function, and activity of daily living (ADL) had been successfully obtained.

When comparing gait ability and hip function at pre- and post-HWA, ROM of hip extension, hip abduction torque, hip extension torque, TUG, gait speed at SWS, cadence at SWS, the WOMAC-p score, and the WOMAC-f score were improved. Hip abduction torque, hip extension torque, TUG, gait speed at SWS, and the WOMAC-p score at post-HWA showed almost equivalent levels at follow-up. According to the weekly measurement results, hip abduction torque, extension torque, gait speed, and TUG were improved quite early. In particular, gait speed at SWS and TUG reached their peak at 3 weeks after THA, and were of almost similar levels in age- and gender-matched elderly people [16]. A previous study suggested that, although walking without assistance is possible during the early post-THA stage, it takes about six months after surgery to improve gait speed and symmetry [6]. In addition, the postoperative walking function has improved with the advancement of surgery and rehabilitation interventions; however, the results of a recent study showed that gait speed did not improve at 4 weeks after THA compared to normal levels in female patients [4]. Our results suggest that gait training with HWA is effective for the early improvement of gait and hip function in patients who have undergone THA. Reduced medical costs can also be anticipated, as early recovery of gait and hip function, and the consequent improved patient independence, would reduce hospital stays and the nursing care burden.

Regarding the kinematics and iEMG during walking, our patient lacked hip extension and had abnormal gait patterns, excessive anterior pelvic tilt, decreased hip ROM, and excessive GMax and GMed amplitude preoperatively. ROM of the hip temporarily worsened as the leg length grew 1 week after THA, which worsened the maximum hip extension and anterior pelvic tilt angle at pre-HWA. However, comparing pre- and post-HWA, maximum hip extension, anterior pelvic tilt angle, and hip range during gait were improved. Furthermore, the iEMG showed a decreased amplitude of GMax, GMed, and GCm. Patients with hip OA showed significantly greater GMed muscle amplitudes and altered muscle activity because of muscle strength loss during walking [17]. These results suggest that HWA can encourage hip extension movement in the stance phase, resulting in an increased maximum hip joint angle during walking and increased hip abductor and extensor torque during walking exercises. These factors contributed to decreasing the effort required from GMax, GMed, and GCm in the stance phase, thereby enabling forward movements by improving the efficiency of muscle output. It can be hypothesized that these kinematic changes encouraged the recovery of the patient’s gait ability.

According to our hypothesis, HWA was effective for our patient who had undergone THA, because she had no paralysis, did not require strong assistance, and needed hip joint movement improvement. With walking exercises using HWA, walking speed, hip joint torque, and TUG were improved early after THA. However, the gait kinematic parameters at 10 weeks after THA in our case did not return to normal levels [8]. It has been reported that improving gait kinetic and kinematic parameters is difficult [6,8,9,10,11]. In our case, although certain effects by HWA were observed, maximum hip extension and anterior pelvic tilt angle at follow-up worsened compared to pre-HWA levels, suggesting that intervention frequency and period should also be considered.

This study has some limitations. First, this work is a single-case study, there was no control group, and the follow-up period was only 10 weeks after THA. However, this case report is necessary for a future comparative study to clarify the safety, feasibility, and determination of evaluation items of gait training with HWA. Moreover, because HWA was used to recover walking ability and hip function early on after THA, we only evaluated the patient for 10 weeks. Second, some gait kinematic parameters did not return to normal levels compared to those of healthy elderly people. In the future, we will conduct comparative studies with a control group, taking into consideration the intervention frequency and duration.

## 5. Conclusions

In this case, gait training with HWA was safe and feasible, and can be considered as an effective intervention for early improvement of gait ability and hip function of patients who had undergone THA with no adverse events. It is also suggested that HWA contributes to the improvement of the gait pattern. In the future, we will conduct a comparative study with a control group, taking into consideration the intervention frequency and duration.

## Figures and Tables

**Figure 1 medicina-55-00069-f001:**
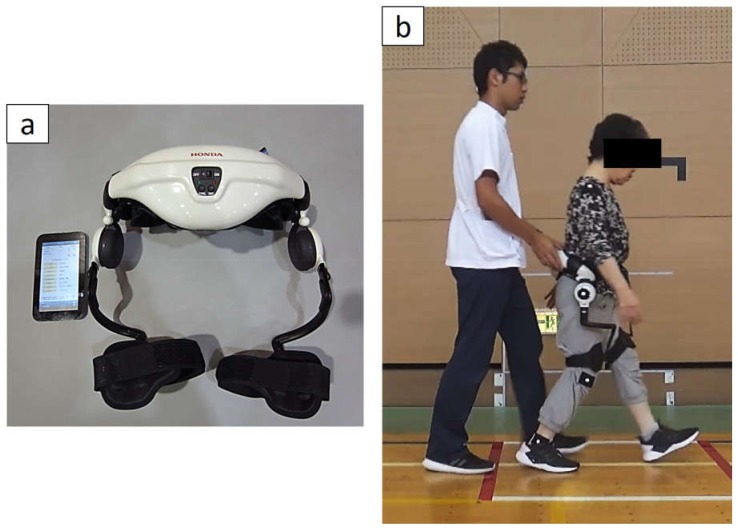
(**a**) The Honda Walking Assistive Device^®^ (HWA) and tablet. (**b**) Gait training with HWA. Motors on the hip joint assist leg movement based on data obtained by the hip angle sensors during walking to improve gait symmetry, lifting a leg from the ground, extending forward during the swing phase, and hip extension to proceed forward in the stance phase.

**Figure 2 medicina-55-00069-f002:**
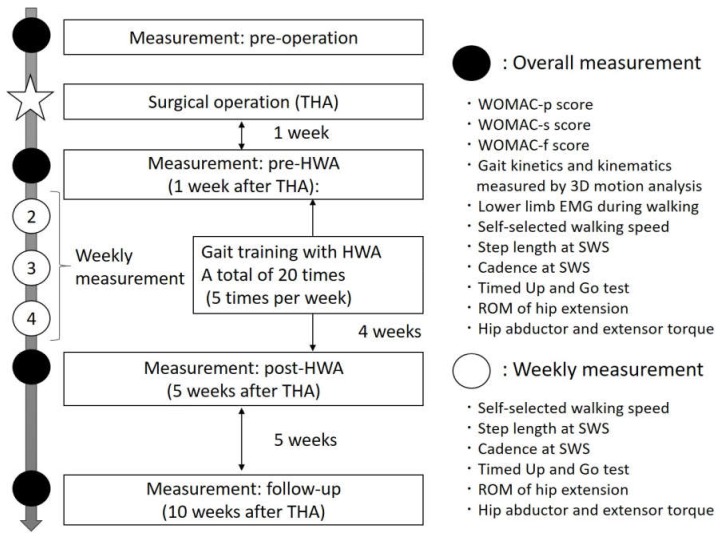
The training protocol: intervention and measurement timing. THA, total hip arthroplasty; 3D, three-dimensional; EMG, electromyogram; SWS, self-selected walking speed; ROM, range of motion; WOMAC, Western Ontario and McMaster Universities Osteoarthritis Index.

**Figure 3 medicina-55-00069-f003:**
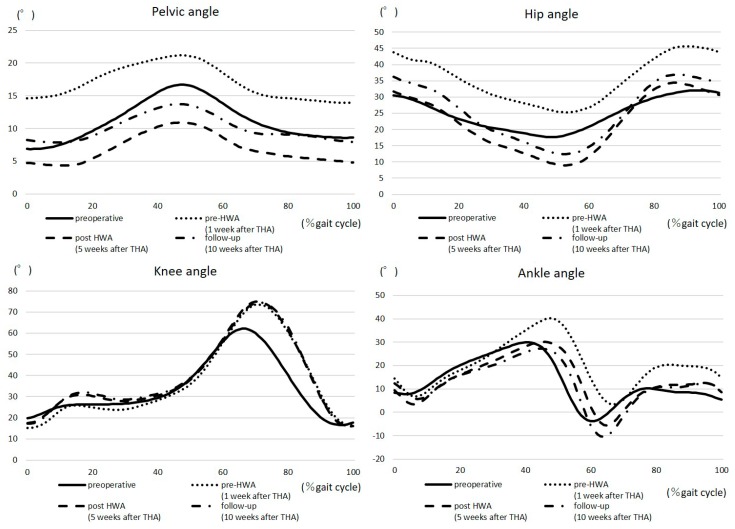
The gait kinematic range at the self-selected walking speed (operative side).

**Figure 4 medicina-55-00069-f004:**
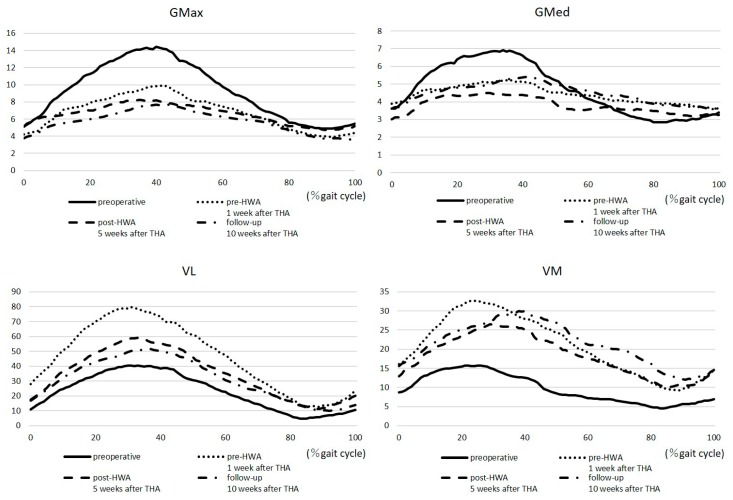
The integrated electromyogram at the self-selected walking speed (operative side). VL, vastus lateralis; VM, vastus medialis; GMax, gluteus maximus; GMed, gluteus medius BF, biceps femoris; TA, tibial anterior; GCm, gastrocnemius medial head.

**Table 1 medicina-55-00069-t001:** The results of measurements related to gait function and hip function.

		Preoperative	Pre-HWA				Post-HWA	Follow-Up	
			1 Week	2 Weeks	3 Weeks	4 Weeks	5 Weeks	10 Weeks	After THA
ROM of hip extension at the operative side	(°)	−10	−25	−15	−10	−10	−5	0	
Hip abductor torque at the operative side	(Nm/kg)	0.35	0.68	0.79	0.76	0.65	0.92	0.80	
Hip abductor torque at the non-operative side	(Nm/kg)	0.52	0.91	0.82	0.85	0.76	0.88	0.96	
Hip extensor torque at the operative side	(Nm/kg)	0.81	0.93	1.55	1.60	1.34	1.56	1.59	
Hip extensor torque at the non-operative side	(Nm/kg)	0.82	1.17	1.37	1.68	1.70	1.59	1.62	
Timed Up and Go (TUG) test	(s)	11.3	12.7	9.9	9.7	9.7	9.6	9.7	
Gait speed at SWS	(m/s)	1.05	1.20	1.31	1.35	1.31	1.36	1.33	
Step length at SWS	(m)	0.50	0.58	0.58	0.59	0.59	0.61	0.56	
Cadence at SWS	(step/min)	126.1	124.7	136.6	138.0	134.1	133.0	143.9	
WOMAC-p		10	5				0	0	
WOMAC-s		5	2				2	3	
WOMAC-f		42	28				19	5	

HWA, The Honda Walking Assistive Device^®^; THA, total hip arthroplasty; ROM, range of motion; SWS, self-selected walking speed; McMaster Universities Osteoarthritis Index subscales of pain (WOMAC-p), stiffness (WOMAC-s), and physical function (WOMAC-f) scores.

**Table 2 medicina-55-00069-t002:** The results on kinematic parameters during gait (operative side).

(°)	Normal [8]	Preoperative	Pre-HWA	Post-HWA	Follow-Up	
			1 Week	5 Weeks	10 Weeks	After THA
Maximum anterior pelvic tilt	9.2	16.7	21.2	10.9	13.7	
Minimum anterior pelvic tilt	4.4	6.9	13.9	4.3	7.9	
Pelvic tilt range	4.8	9.8	7.3	6.6	5.9	
Maximum hip flexion	33.4	30.4	45.7	34.4	36.9	
Maximum hip extension in stance	9.0	−17.7	−25.3	−8.9	−12.4	
Hip range	42.4	12.8	20.4	25.4	24.5	
Maximum knee flexion	64.6	62.1	73.6	75.0	74.5	
Maximum knee extension	0.8	−16.5	−15.2	−15.8	−16.2	
Knee range	65.4	45.6	58.5	59.2	58.3	
Maximum ankle dorsiflexion	13.9	3.8	-3.4	5.6	10.4	
Maximum ankle plantarflexion	24.3	29.8	40.2	30.1	27.2	
Ankle range	38.1	33.6	36.8	35.7	37.6

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
