# Peer review of "Gait Training Using the Honda Walking Assistive Device^®^ in a Patient Who Underwent Total Hip Arthroplasty: A Single-Subject Study"

_medicina, 2019, doi:10.3390/medicina55030069_

Round 1

Reviewer 1 Report

There are some grammatical errors in Abstract. Except that, please accept in present form.

Author Response

Thank you for your comments. As you kindly suggested, we improved the abstract. In the manuscript, colored words are revised parts according to your suggestions.

Reviewer 2 Report

I hope you can continue the study to go beyond the case study

Author Response

Thank you for your comment. We really appreciate your careful reading our manuscript.

This manuscript is a resubmission of an earlier submission. The following is a list of the peer review reports and author responses from that submission.

Round 1

Reviewer 1 Report

This is an analysis using a new robotic assist device for rehabilitation after total hip arthroplasty. This manuscript was well written and the concept was new; however, this evaluation was performed for only one patient. That is the biggest drawback of this paper. I strongly recommend that authors should recruit another several patients with HWA and without HWA use. The comparison of performance between before and after HWA use could be regarded meaningless because readers may think patients could improve their gait and posture after THA even without an assist device.

Minor grammatical errors;

line 90

were performed for less than 20 minutes?

line 154 

increase?

Reviewer 2 Report

the work was carried out correctly but as you pointed out in the conclusions it is poor of of scientific value because it is related to a single case and without a control group.

For a future study I suggest to include a follow up to 6 months of the operation.

It was not mentioned if the patient used crutches during rehabilitation.

I suggest to complete the study with at least one control patients